# Use of Intersectionality Theory in Interventional Health Research in High-Income Countries: A Scoping Review

**DOI:** 10.3390/ijerph20146370

**Published:** 2023-07-15

**Authors:** Laura Tinner, Daniel Holman, Stephanie Ejegi-Memeh, Anthony A. Laverty

**Affiliations:** 1Population Health Sciences, Bristol Medical School, University of Bristol, Bristol BS8 1UD, UK; 2Department of Sociological Studies, The University of Sheffield, The Wave, 2 Whitham Road, Sheffield S10 2AH, UK; 3Public Health Policy Evaluation, School of Public Health, Imperial College London, London SW7 2BX, UK

**Keywords:** intersectionality, health inequalities, health interventions, evidence synthesis

## Abstract

Background: Intersectionality theory posits that considering a single axis of inequality is limited and that considering (dis)advantage on multiple axes simultaneously is needed. The extent to which intersectionality has been used within interventional health research has not been systematically examined. This scoping review aimed to map out the use of intersectionality. It explores the use of intersectionality when designing and implementing public health interventions, or when analysing the impact of these interventions. Methods: We undertook systematic searches of Medline and Scopus from inception through June 2021, with key search terms including “intersectionality”, “interventions” and “public health”. References were screened and those using intersectionality and primary data from high-income countries were included and relevant data synthesised. Results: After screening 2108 studies, we included 12 studies. Six studies were qualitative and focused on alcohol and substance abuse (two studies), mental health (two studies), general health promotion (one study) and housing interventions (one study). The three quantitative studies examined mental health (two studies) and smoking cessation (one study), while the three mixed-method studies examined mental health (two studies) and sexual exploitation (one study). Intersectionality was used primarily to analyse intervention effects (eight studies), but also for intervention design (three studies), and one study used it for both design and analysis. Ethnicity and gender were the most commonly included axes of inequality (11 studies), followed by socio-economic position (10 studies). Four studies included consideration of LGBTQ+ and only one considered physical disability. Intersectional frameworks were used by studies to formulate specific questions and assess differences in outcomes by intersectional markers of identity. Analytical studies also recommended intersectionality approaches to improve future treatments and to structure interventions to focus on power and structural dynamics. Conclusions: Intersectionality theory is not yet commonly used in interventional health research, in either design or analysis. Conditions such as mental health have more studies using intersectionality, while studies considering LGBTQ+ and physical disability as axes of inequality are particularly sparse. The lack of studies in our review suggests that theoretical and methodological advancements need to be made in order to increase engagement with intersectionality in interventional health.

## 1. Introduction

The Marmot Review 2020 found that there has not been the expected reduction in health inequalities in the UK and some inequalities have widened [1]. This picture is similar across many other high-income countries, with strategies having limited success in dealing with the long term consequences of the financial crisis of 2008, among other structural issues [2].

These difficulties have given rise to new forms of framing health inequalities and an increased desire for strategies to overcome identified barriers and help to theorise and communicate their nature, causes and solutions [3]. Intersectionality explicitly considers multiple axes which may give rise to health inequalities and represents a potentially promising way forward on these issues [4]. The concept was first developed by Crenshaw in 1980 and is rooted in Black feminist and critical legal theory [5]. It is based on the premise that there are multiple social forces, social identities and ideological instruments through which power and disadvantage are expressed and legitimized [5]. While there is no clear consensus about the exact definition of intersectionality, a key focus on social justice and mutually cross-cutting and interacting dimensions of identity are commonly used [6,7]. Core principles include: (1) an explicit focus on structural factors or social determinants of health; (2) consideration of discrimination, particularly across more than one axis of inequality; (3) a focus on an equitable power dynamic with communities and users of services [8].

Intersectionality has been adopted in research to consider and assess ‘intersections’ of a wide range of identities and positions, including ethnicity, socioeconomic position, gender, LGBTQI+ and physical disability [9]. Intersectionality has been described as an essential theoretical framework that may be particularly useful to address health issues and their impact on the most vulnerable populations [6]. There are a range of analytic and theoretical frameworks that come under the concept of intersectionality, with substantial heterogeneity between these. Prominent examples such as the Intersectionality-Based Policy Analysis (IBPA) attempt to set out policy-based frameworks that support the decision-making processes among stakeholders working in health-related sectors [7,10]. This framework has provided a structure to critically analyse policy, to understand policy contexts and to generate innovative equity-focused policy solutions. Other frameworks aim to offer a context for conceptualising and interpreting intersectionality at the individual level for both quantitative and qualitative research [11]. However, despite the strong theoretical rationale for adopting intersectionality within public health research and a suite of developed frameworks for researchers to employ it, little is known about whether intersectionality (either these noted frameworks or other manifestations) is being translated into health-related interventional research. Mapping out the extent of intersectionality’s use within the field will help ascertain whether researchers are adopting this potentially useful concept into their applied research, or whether intersectionality may remain stagnant as a purely theoretical concept. Scoping out the current evidence base will also identify successful (or unsuccessful) adoption of intersectionality, to provide learning for other researchers developing or evaluating interventions. Until we have an idea of the extent of use, we are unable to make assessments of its interventional utility and make recommendations for implementation.

Within public health, the possibility of interventions exacerbating health inequalities has been highlighted and is beginning to be investigated through subgroup analyses [12,13]. However, interventions which focus on only one axis of inequality are similarly limited and risk worsening inequalities by not sufficiently tailoring delivery to different groups or evaluating the effects to account for interactions. Intersectionality could thus prove useful in designing and evaluating public health interventions, allowing researchers to determine the equitability of their intervention in a way that accounts for multiple interlocking inequalities, yet review evidence is lacking and the extent to which it is being employed is relatively unknown. One systematic review published in 2021 mapped the presence of intersectionality in quantitative health research [14]. However, this study did not explore the use of intersectionality in studies of health interventions, a gap we intend to address. Further, an overview by Heard et al. [15] provides some direction around how intersectionality is beginning to be incorporated into public health promotion and gives examples related specifically to public health interventions. For instance, consulting people from minority ethnic backgrounds at the start of intervention development to recognise alternative consequences for people who sit at unique social positions. Crucially for our review, Heard et al. reiterate that applying intersectionality to methodology remains under-explored, but there are innovative modelling methods emerging which may advance quantitative intersectionality-informed analysis [15,16].

Given this potential utility of intersectionality for interventional research, our scoping review seeks to clarify how intersectionality is being used within interventions through systematic scoping methods. It is hoped that, through doing this, we will gain a better idea of the landscape of interventions using intersectionality and can begin to identify successes, challenges and recommendations to help researchers design more equitable interventions. Our review provides an initial step in this process, through examining the extent to which intersectionality frameworks are used within interventional health research.

### Review Aim

Our review aim was: To explore what evidence is there on the use of theoretical and analytical intersectionality frameworks when designing and analysing public health interventions. Our scoping review set out to explore the use of intersectionality theory and/or frameworks when designing or implementing public health interventions. We also aimed to identify intersectionality-based analytical approaches to examining the impact of interventions on health inequalities. 

## 2. Methods

Given the nature of our aim, we used a scoping review to allow flexibility to capture a range of intersectionality frameworks. Scoping reviews summarise evidence to convey the breadth and depth of a field, and may involve some analytical reinterpretation of the research literature [17,18]. We followed the Preferred Reporting Items for Systematic reviews and Meta-Analysis extension for Scoping Reviews (PRISMA-ScR) reporting guidelines and align with recommended methods for scoping reviews [17,19,20,21].

We used a configurative approach to identify intersectionality frameworks supporting interventions. Configurative syntheses focus on gathering evidence to elucidate certain processes rather than testing any intervention effects. We thus aimed to organise available data to better understand and answer our review question.

### 2.1. Eligibility

We included studies exploring the utility of intersectionality frameworks either to design/implement or to analyse the impacts of public health interventions. We only included interventional health-related studies that aimed to effect physical or mental health outcomes. In line with the core concepts of intersectionality, our health inequality search terms included ethnicity, socioeconomic position, gender, LGBTQI+ and disability. We included studies which directly mentioned intersectionality theoretical or analytical frameworks within the paper.

#### 2.1.1. Inclusion Criteria 

To be included in our review, studies should address intersectionality when designing, implementing and/or evaluating interventions. We remained open to a range of uses of intersectionality, expecting heterogeneity between studies in both the degree to which it was employed as either a framework for design or analysis, as well as the way it was interpreted and applied. Therefore, there may be studies that mention that they use intersectionality as a lens in their early discussions during the design of the intervention as well as studies that employ intersectionality as a guiding framework throughout the whole of the design and evaluation process—and both these would be included under our criteria. All study types were accepted, including quantitative, qualitative and mixed-method approaches. 

We limited our inclusion criteria to high income countries for a few reasons. First, the intervention implementation context between high income countries (HICs) and low-and-middle income countries (LMICs) can be quite different, and reviews regularly separate inclusion criteria in this way. While intersectionality is a potentially useful concept for both contexts, the systems of power and oppression are likely different in different contexts, an idea beginning to be explored through applying intersectionality with the consideration of geographic and socioeconomic contexts [22]. One article highlights the second reason for reviewing HICs separate from LMICs, in that studies employing intersectionality in LMICs tend to be focused on immunization, HIV and violence and sexual abuse [23]—all public health topics quite different to health behaviour and mental health topics typical of HIC interventions. Finally, the recent systematic review on intersectionality in public health research cited in the background did include studies from both high and low-and-middle income countries and found that the vast majority of studies were conducted in high income countries [24]. Therefore, while our scoping review is limited by this omission, we do not expect there would have been many additional studies to include, yet a considerable amount more studies to screen.

To be included, we restricted studies to being health-related studies (i.e., measuring or analysing any indicator or topic related to physical or mental health of populations, as well as health system related outcomes), as opposed to educational or social outcomes, providing original results. 

#### 2.1.2. Exclusion Criteria

We excluded studies that were commentaries, editorials, book reviews or studies exclusively focused on educational, sociological or judicial issues. 

### 2.2. Search Strategy

The main search strategy was conducted in Medline and Scopus in June 2021. The search strategy was initially designed for Medline and then adapted to be replicated in Scopus. The search terms used were related to “intersectionality”, “interventions” and “public health” (Appendix A for full search strategy). The searches were not limited by year or language. In addition, the citations of identified key papers were screened to find relevant studies that had not been captured by the search strategy. The retrieved references were stored and managed using EndNote X9 (Clarivate, Emeryville/Berkley, CA, USA). References were exported to the web-based software Covidence (Veritas Health Innovation, Melbourne, Australia) for screening.

### 2.3. Screening 

The screening was conducted by two researchers (ARG and LT). Initially, the titles and abstracts of articles identified from the search strategy were screened against the inclusion and exclusion criteria. Then, the same process was carried out with the full text of selected articles. Any disagreement was solved through discussion with a third researcher (AAL). 

### 2.4. Data Extraction

A data extraction form was created including the following fields: study design, setting, number of participants, year of publication, country, target population, inequalities targeted, areas of intervention, summary of quantitative and qualitative outcomes, description of intersectionality framework, intersectionality approach and any additional notes.

### 2.5. Synthesis of the Results 

Research evidence from quantitative, qualitative and mixed-method studies was summarised narratively. We described studies according to their target populations and axes of inequality addressed. Studies were categorised into those where intersectionality frameworks were used (1) in order to design their interventions from the outset, and (2) after interventions were implemented in order to analyse effects.

### 2.6. Selection Process

The search strategy retrieved 2568 references, of which 460 were duplicates (Figure 1). The titles and abstracts of 2108 references were screened, resulting in 383 references included in full text screening. The most common reasons for exclusion were “studies not using any intersectionality framework” (193 studies) and “studies not providing original results” (121 studies). Thirty-two studies were excluded as they did not contain any health-related outcomes and exclusively focused on educational, sociological and judicial outcomes. Twelve studies met the inclusion criteria and were included in the review.

## 3. Results 

### 3.1. Characteristics of Included Studies

Of the twelve included studies, six studies were based in the USA, four studies were conducted in the UK, one in Canada and one across several countries (USA, UK, Australia, New Zealand and Norway) (Table 1). All studies were published between 2014 and 2021. Six studies were qualitative and focused on alcohol and substance abuse (two studies), mental health (two studies), general health promotion (one study) and housing interventions (one study). 

The three quantitative studies examined mental health (two studies) and smoking cessation (one study), while the three mixed-method studies examined mental health (two studies) and sexual exploitation (one study). Full details of included studies are in Table A2. 

There was substantial heterogeneity in terms of the target populations. Included populations were: health researchers [25]; Polish migrants [26]; mental health service users [27]; women on (or previously on) opioid substitution treatment [28]; single adults in receipt of welfare for housing [29]; pregnant/postpartum women [30]; Asian men affected by mental illness [31]; people who smoke tobacco [32]; people receiving individual mental health counselling [33]; youth at risk of sexual exploitation [34]; homeless women [35]; Latina women who had experienced interpersonal violence [36]. The number of participants varied between 17 and 415, with lower sample sizes in the qualitative studies.

The most commonly addressed individual axes of inequality were ethnicity and gender, which were considered by 11 studies each (Table 2 and Figure 2). SES was considered by 10 studies. LGBTQ+ was considered by four studies and disability by one study. Of our five pre-defined axes of inequality, only one study considered all five, with two studies considering four of these axes. Six out of the twelve studies considered three out of five of our pre-defined axes of inequality. There was heterogeneity regarding which intersections (e.g., sex X ethnicity X age), although SES, ethnicity and gender were considered by eight out of the twelve included studies. 

Eight studies used intersectionality to analyse the association between inequality axes and intervention outcomes [26,27,28,29,31,32,33,35]. The role of intersectionality within these studies was mainly related to the interpretation of the findings in terms of cross-cutting inequalities. Four of these studies analysed mental health related problems [27,31,33,35], one study focused on smoking cessation [32], one on housing [29], one on alcohol abuse and one on drug abuse [26,28]. There were considerable differences in the use of intersectionality. Whilst, in some studies, intersectionality clearly structured the analysis of the results [28,31,32], other papers only used intersectionality as one option for the interpretation of the findings [27].

Three studies utilised intersectionality to design interventions with an aim to enhance their effectiveness on reducing health inequalities [25,34,36]. In these studies, intersectionality was consistently used to guide the design of the interventions across the studies. One study used intersectionality for both the design of the intervention as well as the interpretation of findings [30] (Table 2). 

#### 3.1.1. Use of Intersectionality to Analyse Impacts of Interventions (Eight Studies)

Four of the studies which used intersectionality to analyse the effects of interventions were on mental health. These studies examined various intersections using gender, ethnicity, SES and LGBTQ+ indicators. In addition to analysis of intervention effects, one study used intersectionality to decide what axis of inequality should be used to measure effects [33]. The other three studies concluded their analyses by proposing intersectionality as a useful framework for improving or tailoring services in the future. These studies demonstrate an awareness and use of the key tenets of intersectionality: three of them use such frameworks from the outset, including Kivlighan et al. [33], which addresses the question of whether intersecting identities effect treatment outcomes. All of these studies conclude that the use of intersectional frameworks is beneficial in maintaining suitable awareness of social dynamics and recommend their further use. In one study by Lloyd et al. [27], issues of intersectionality were raised by the participants as a way to improve treatments. 

Of these studies on mental health, Lloyd et al. [27] interviewed and surveyed LGBQ+ patients who had completed a cognitive behavioural programme to overcome mental health issues specifically designed for sexual minorities. An important suggestion for developing and improving this therapy from the patients was the use of an intersectional lens to acknowledge the totality of people’s experiences and to move beyond a predominant focus on male sexuality and identity. Kivlighan et al. [33] used an intersectionality framework to investigate therapist–client interaction in relation to gender and ethnicity. Their analysis of 415 clients treated by 16 therapists found that there were differences in therapists’ ability to reduce psychological distress depending on the intersection of ethnicity and gender. Morrow et al. [31] used intersectionality as an analytical framework to explore Asian men’s diverse experiences of stigma and mental health, including how this is mediated by a range of other identities and experiences of racism, inequality and immigration. They report that participants understood and experienced their stigma as linked to their social identities and that intersectional frameworks must feature prominently in attempts to reduce mental health stigma. David et al. [35] used an intersectional framework in their mixed-method evaluation of mental health services delivery for women who are homeless. Their findings highlight the importance of tailoring treatments to account for the multiple forms of oppression faced by these women and assuming a culturally sensitive therapeutic stance in treatment. They specifically reference the intersectional oppression faced by these homeless women and advocate the use of an intersectional perspective in providing treatment.

The remaining four studies using intersectionality to analyse the impacts of interventions targeted smoking cessation, housing and alcohol and drug misuse. These studies all used intersectionality frameworks to assess whether the interplay of social identities is more important to consider than individual axes of inequality. Three of these studies examined whether outcomes differed according to intersectional markers of identity, and one recommended the consideration of intersectionality as a method to improve services. They did not, however, contain explicit reference to power dynamics or discrimination. 

Potter et al. [32] used an intersectional framework to assess how the interplay of multiple marginalised attributes contributes to smoking cessation. This study found that low household income was related to continued smoking but identified no interaction between marginalised attributes and relapse. Gleeson et al. [26] explored associations between social attitudes towards gender and access to services related to migrant status and social class according. Their findings suggested that Polish female migrants accessing alcohol-related services face barriers including social stigma and sexist attitudes towards women, and that these interact to create negative outcomes. Medina-Perucha et al. [28] studied the intersection of different types of inequalities in women receiving opioid substitution treatment with a focus on issues of stigma and discrimination. They highlighted that stigma interacts with other aspects of the women’s identities, including drug use and homelessness, to create negative outcomes. Wilkinson and Ortega-Alcázar [29] studied the impact of shared housing on young people and their wellbeing using an intersectional perspective. Their findings stressed the importance of considering the intersection of different types of inequalities on physical safety and violence, mental health and isolation.

#### 3.1.2. Intersectionality as a Tool to Design Interventions (Four Studies)

Three studies used intersectionality to design interventions [25,34,36]. We have included Stevens et al. [30] in this category, which used intersectionality for both design and analysis. The main value of intersectionality in these studies was to provide the appropriate mechanisms to culturally adapt interventions, in order to cover the intersection between different types of health inequalities. In these studies, intersectionality was specifically used to address social factors related to ethnicity and gender [25,30,36], and to a lesser extent socioeconomic position and LGBTQI+ [34], within intervention design. In general, these studies applied key tenets from intersectionality frameworks with a focus on structural factors and power dynamics. 

Liu et al. interviewed 26 health researchers and promoters to explore their views on adapting interventions for ethnic minority women [25]. Researchers interviewed in this study were clear that intersectionality was helpful in understanding the combined influence of ethnicity and gender among other factors in health promotion interventions. They highlight that intersectionality, as well as the concepts of representation and contextual experiences, are all useful in understanding how adapting interventions works in practice. Bounds et al. [34] and Kelly and Pich [36] both used intersectionality frameworks to acknowledge and address issues of power and discrimination, in line with the key tenets of intersectionality. Bounds et al. [34] aimed to adapt an intervention for newly homeless youth to be used to reduce risk factors for sexual exploitation. Participants recognised the importance of acknowledging experiences of structural violence, while the authors concluded that focus groups and intersectionality are useful in considering the unique issues of disempowered youth. Kelly and Pich [36] aimed to reduce post-traumatic stress disorder and improve quality of life, social support and self-efficacy among Latinas who experienced intimate partner violence. The authors integrated both biomedical and intersectional approaches throughout their study, adapting a family intervention to reduce risk factors for sexual exploitation and acknowledging issues of power and invisibility. They concluded that that interventions helped to improve mental health related symptoms, with an intersectional approach being key to this, although there were limited impacts on other outcomes. 

The only study which used intersectionality to design and analyse their intervention was Stevens et al. [30]. This study used intersectionality to assess the effectiveness of a coordinated perinatal mental health care model focused on socially disadvantaged ethnic minority women. They use the framework to conceptualise the ‘vulnerability’ of these women as influenced by a range of interacting structural factors. They found similar treatment outcomes among ethnic minorities as in White patients and concluded that their treatment model based on intersectionality has promise to reduce inequalities in this area, over and above approaches without an intersectionality framework. 

## 4. Discussion

This scoping review aimed to explore the use of intersectionality in the design and/or assessment of interventional health research. Our findings suggest that intersectionality frameworks are not yet explicitly used in this body of research, as we identified only 12 studies in total. In the studies we identified, the axes of inequality gender, ethnicity and socio-economic position were commonly assessed, but there was less attention paid to other markers of inequality. In our included studies, intersectionality frameworks were used to pose questions about intervention effectiveness and interpret differences in outcomes, and were recommended for use in tailoring and delivering future treatments. We found that intersectionality frameworks were more commonly used to assess the impacts of pre-existing interventions, rather than to use these frameworks to develop new methods of tackling health inequalities. 

To the best of our knowledge, this is the first review studying the use of intersectionality frameworks within interventional health research. Some previous research has explored the association between intersectionality and health through theoretical, qualitative and observational studies, without considering health interventions specifically [14]. One previous review analysed evidence on the use of intersectionality in health by mapping its presence [14]. This review pointed to significant room for improvement in explicitly connecting research methods and reporting to intersectionality frameworks in studies and focusing more on interventions. Our scoping review builds on this work and the fact that we found so few studies suggests there is still some way to go before intersectionality is comprehensively incorporated into interventional research. 

Other work by Harari and Lee has identified limitations in quantitative research using intersectionality, including the prioritisation of certain groups and not others, and a lack of consideration of underlying processes [37]. While we found only a limited number of studies, those quantitative studies which we did identify chime with Harari and Lee’s assessments. Promisingly, however, there was also evidence that the included studies had taken care to consider diverse identities and drivers of inequality, including stigma, power dynamics and identity. However, some characteristics were included more commonly than others: we found ethnicity and gender to be the most commonly included intersections, but a comparative lack of consideration of LGBTQi+ and disability. This, perhaps, is unsurprising given that intersectionality was borne out of an interest specifically in the position of Black women, and that gender and ethnicity are common demographic characteristics collected in intervention research [5,38]. 

Researchers have warned against intersectionality work sliding into ‘oppression Olympics’ [39], in which we try to determine which characteristic is most important for a health outcome instead of appreciating the nuance and complexity of people’s intersectional positions. Our finding that gender and ethnicity were the most commonly included characteristics, while expected given the recognition of these as key dimensions of inequality and thus greater data availability, reflects the debates around differentiating categories within intersectionality [40]. There are three approaches to this: intra-categorical approaches that focus on complexity of experience within a particular social position or intersection, inter-categorical approaches that focus on heterogeneity across a range of intersections and anti-categorical approaches that critique rigid social categorization itself [14]. There is an argument that interventional research needs to categorise to some extent to determine the success of the strategy, with gender and ethnicity being core categories by which researchers want to understand intervention effects. Interventional research has therefore mostly used inter-categorical approaches, working within a traditionally positivist paradigm [15]. To begin to address this challenge and to uncover inequalities in a more nuanced way, there is a need for innovative and mixed methods that can explore the richness and detail of people’s experiences while also conducting intervention effect comparisons. 

The studies included used a range of methods to employ intersectionality, drawing on different concepts dependent on their methods. Although intersectionality is increasingly highlighted as a promising framework for public health research, there is still uncertainty and challenge about how to operationalise it methodologically [3]. It is therefore unclear how to determine whether intersectionality was appropriately applied to the different methods used by the included studies. What we can say is that all the included studies used intersectionality as an analytic framework, rather than approaching the presence or absence of intersectionality as a testable hypothesis [41]. This finding is positive, given calls for intersectionality to be used analytically, rather than purely descriptively. Regarding topic area, we found that mental health accounted for half of the included studies, which could be related to the burgeoning field of research that has identified social inequalities in mental health across multiple dimensions of inequality, such as socioeconomic position, gender and sexual orientation, among others [42]. Further, within mental health research, there has been increasing focus on how these dimensions of inequality may need to be incorporated into analysis to identify complex and potentially unexpected patterns in the distribution of health [42]. Finally, mental health research is well suited to intersectionality, as the people and groups occupying positions of multiple disadvantages may well experience identity issues, stigma, discrimination and disempowerment.

We found limited evidence on the explicit use of intersectionality frameworks to support the development of interventions. Those studies which did, however, had a strong recognition of the issues of power, structural dynamics and discrimination, and they considered these in intervention development. There is an inherent complexity in addressing these issues, which may explain in part why we found a limited number of studies. Furthermore, some of these components have been recognised as useful elements to tackle inequalities without being mentioned as part of any intersectionality framework [43]. Hence, rather than being underrepresented, intersectionality might be to some extent underreported, since many of its main components may be currently used in health interventions, even though they are not framed as intersectionality in any theoretical background. 

The benefits of incorporating intersectionality into public health research are well-established in theory and have been championed by scholars over the past few decades [5,10,11,14]. What our scoping review intended to contribute was evidence as to whether intersectionality is being actively used by researchers in the development and evaluation of interventions. What we found was limited use of intersectionality within interventional research, perhaps as a result of some of the points we have discussed. There is motivation from many within the public health community to adopt intersectionality as a way of reducing health inequalities but translating it into methodological approaches remains a challenge. While our small sample size limits what we can recommend in terms of intersectionality and intervention research, we would suggest that incorporating intersectionality in the design of interventions (not as a post-intervention analysis), would strengthen the equity focus. As fewer studies used intersectionality in this way, our review conveys that there is still some way to go before this becomes more commonplace. We would direct researchers to resources such as Hankivsky et al.’s [44] primer and Heard et al.’s paper [15], which both provide case study examples of innovation using intersectionality frameworks in public health research. Additionally, the ‘For-Equity’ website also provides tools and resources related to intersecting inequalities to enable researchers to think about this when developing and evaluating interventions [45]. 

### Strengths and Limitations 

This review presents a timely review of research on the use of intersectionality in interventional health research. It identifies that intersectionality is not yet commonly used in this body of research. The reduction of health inequalities is a core component of public health and has been since its inception, and so this is a highly relevant topic within public health research. Our most recent search was undertaken in June 2021, so there may be relevant studies missing from our analysis. Staff changes affected our capacity to update the searches. We used a broad, pre-defined comprehensive search strategy using two of the main biomedical databases, and the citations of identified key papers were also screened. Nonetheless, the main limitation of this review is that we only included papers explicitly mentioning intersectionality or related terms. We did pilot searches in an attempt to identify papers which used key tenets of intersectionality. This, however, was not feasible, as almost all papers would have to be read at full text stage in order to assess if they had used intersectional ideas or those of interacting identities in the consideration of their results. This means that we may have missed studies which did not use the term explicitly, but which used strategies in line with intersectionality frameworks. As such, our results likely constitute an underestimate of the extent of use of intersectionality frameworks in interventional health research.

A further limitation of our scoping review is that we limited our inclusion criteria to high income countries only. This means our findings are non-transferrable to low-and-middle income countries. There is work that explores intersectionality use in research in low-and-middle income countries [23], and also a review that does not separate in this way [24]. We would suggest that future work seeks to compare these two intervention contexts, as it may have implications for the extent and ways intersectionality is implemented. 

Our scoping review intended to map out the extent to which intersectionality is being used in interventional health research and highlight some examples of its use. An extension of this work would assess whether and how intersectionality impacts on the quality of the interventional health research comparative to the use of other theoretical and analytical frameworks. Finally, as this was a scoping review focused on exploring the use of intersectionality, we did not conduct formal critical appraisal of included studies.

## 5. Conclusions

In conclusion, this scoping review has revealed there is a lack of evidence on the use of theoretical and analytical intersectionality frameworks when designing and analysing public health interventions, with only a small number of studies identified. Where intersectionality was adopted, frameworks were used to pose questions about intervention effectiveness or interpret differences in outcomes, rather than in the design or implementation of interventions. Therefore, intersectionality within this small sample was primarily used as an analytical framework. Public health research is increasingly highlighting the value of intersectionality frameworks for attempting to reduce inequalities. The lack of studies in our review suggests that theoretical and methodological advancements need to be made in order to enhance engagement with intersectionality as part of the health intervention development and assessment cycle. 

## Figures and Tables

**Figure 1 ijerph-20-06370-f001:**
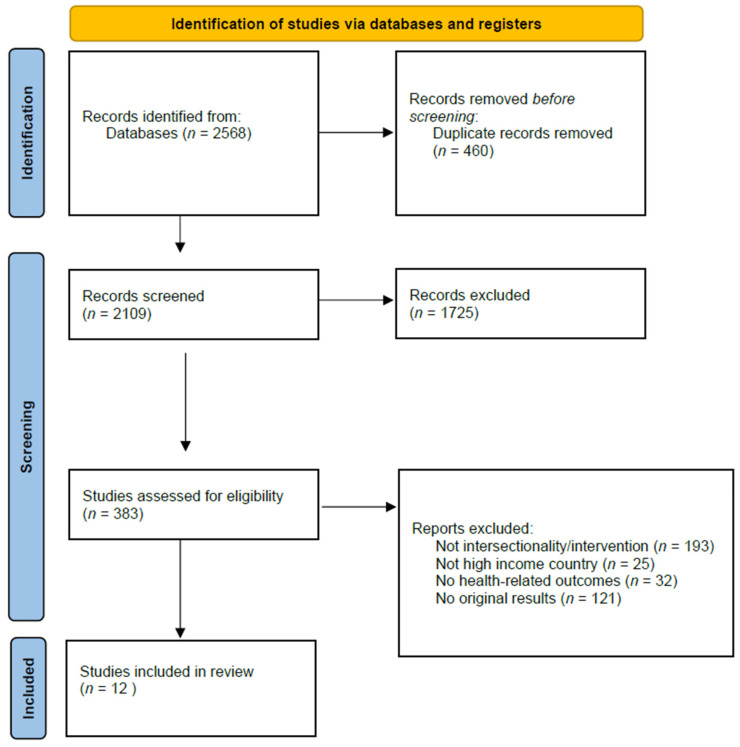
PRISMA Flow Diagram.

**Figure 2 ijerph-20-06370-f002:**
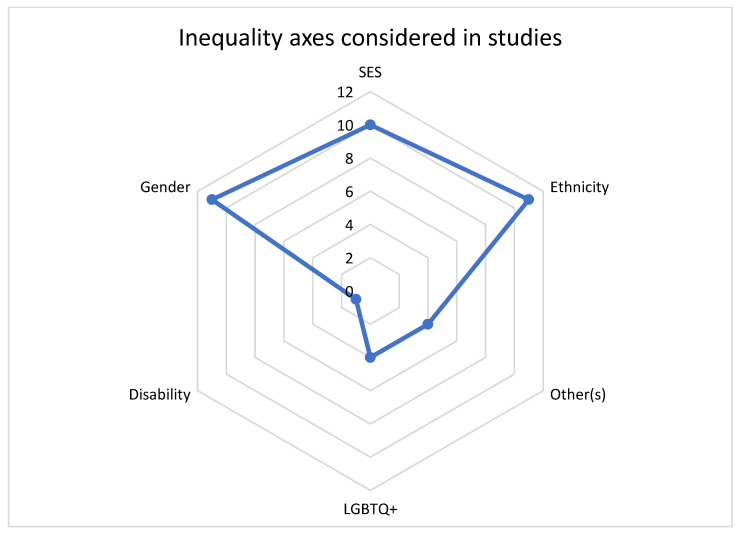
Axes of inequality considered in included studies.

**Table 1 ijerph-20-06370-t001:** Summary of studies features.

	*n* = 12	%
**Country**		
USA	6	50
UK	4	33.33
Canada	1	8.33
Several countries	1	8.33
**Year**		
2021	2	16.66
2020	3	25
2019	3	25
2018	1	8.33
2017	0	0
2016	1	8.33
2015	1	8.33
2014	1	8.33
**Research design**		
Qualitative	6	50
Quantitative	3	25
Mixed-method	3	25
**Intersectionality use**		
Analysis	8	66.66
Design	3	25
Design/Analysis	1	8.33

**Table 2 ijerph-20-06370-t002:** Study characteristics and intersections.

Author(s)	Country	Health Topic	Sample Size	Intersectionality Use	Inequalities					
					SES	Ethnicity	Gender	LGBTQ+	Disability	Other(s)
Gleeson et al. (2020) QUAL [26]	UK	Alcohol misuse	17	Analysis						NA
Liu et al. (2016), QUAL [25]	Various	Health promotion (Various)	37	Design						Age
Lloyd et al. (2021) QUAL [27]	UK	Mental health	18	Analysis						Age
Medina-Perucha et al. (2019) QUAL [28]	UK	Drug abuse	20	Analysis						Various
Wilkinson and Ortega-Alcázar (2019) QUAL [29]	UK	Housing	40	Analysis						NA
Stevens et al. (2018) QUAL [30]	USA	Mental health (perinatal)	82	Design /Analysis						NA
Morrow et al. (2020) MIXED [31]	Canada	Mental health	94	Analysis						Age
Potter et al. (2021) QUANT [32]	USA	Smoking cessation	344	Analysis						NA
Kivlighan et al. (2019) MIXED [33]	USA	Mental health	415	Analysis						NA
Bounds et al. (2020) MIXED [34]	USA	Risk for sexual exploitation	40	Design						NA
David et al. (2015) QUANT [35]	USA	Mental health	300	Analysis						NA
Kelly and Pich (2014) QUANT [36]	USA	Mental health	27	Design						NA
					10	11	11	4	1	4

## Data Availability

All data generated or analysed during this study are included in this published article.

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
