# Peer review of "Use of Intersectionality Theory in Interventional Health Research in High-Income Countries: A Scoping Review"

_ijerph, 2023, doi:10.3390/ijerph20146370_

Round 1

Reviewer 1 Report

The work is good and will contribute to the evidence base of the use of intersectionality in public health interventions. Some thoughts:

- has the research team considered updating the search - June 2021 is almost 2 years ago; do they anticipate a significant difference?

-  please include a rationale for why 'high income countries' was an inclusion criteria

Otherwise, I would recommend the manuscript be published. No recommended changes.

Author Response

Thank you for these useful comments.

The issue of the search timelines is a challenge that has remained in our minds during the writing up process. There were considerable delays to our review from the writing up stage, due to staff changes, sickness and other delays. We have considered re-running the searches, but due to low staff capacity and the complexity of the studies we are unable to do so at this time. While we know that some studies may have completed during this time, interventional research is not a rapidly developing field. These interventions take many years to complete and the vast majority of ongoing studies will have been paused during the pandemic. We hope the reviewer appreciates that this review would have been published a while ago had it not been for these issues outside of our control. While we would update the searches if we had capacity, we do not believe that our findings would be altered as a result because we expect very few additional studies would be included. We have noted this issue in the limitations (Page 13, line 20)  

 Our most recent search was undertaken in June 2021, so there may be relevant studies missing from our analysis.

Please see response to the Editor above, in the third comment row, which addresses this important comment.

Reviewer 2 Report

The review study analyses the application of the theory of intersectionality in health interventions in high-income countries. This is vital, as it is a theoretical proposal that allows us to understand health inequalities in light of the multiple discriminations individuals suffer. This theory makes it possible to explain those combinations of discrimination that would otherwise appear to be isolated and less harmful.

My main question is, why only include studies on rich countries rather than developing countries, where there is supposed to be more inequality? Therefore, I suggest including studies of developing countries or better justify why only rich countries are included.

I suggest turning the research question into a research objective.

I suggest working with the most up-to-date version of PRISMA (2020). https://doi.org/10.1136/bmj.n71

The "Selection process" and "Characteristics of included studies" sections should be in the Methods section, not in Results.

The Conclusions should respond to the general and specific research objectives.

Revise the wording, for example, in line 172, "and and" and in other parts of the text.

Revise the IJERPH citation style; you cite with endnotes but without brackets.

Revise the contraction "et al"

Author Response

Please see response to the Editor above, in the third comment row, which addresses this important comment.

Thank you we have now done this.

We have now input the flow diagram information into the PRISMA (2020).

We have moved the “selection process” to the methods section, thank you for the suggestion. We have kept the “Characteristics of included studies” in the results, as we believe this is conventional for review studies in this journal. If the editorial team wish us to change this, we are happy to.

We have reworded the conclusion to more directly relate to our research objectives of exploring the use of intersectionality in the design and implementation of public health interventions and to identify analytical approaches (page 14, line 2)

In conclusion this scoping review has revealed there is a lack of evidence on the use of theoretical and analytical intersectionality frameworks when designing and analysing public health interventions, with only a small number of studies identified. Where intersectionality was adopted, frameworks were used to pose questions about intervention effectiveness or interpret differences in outcomes rather than in the design or implementation of interventions. Therefore, intersectionality within this small sample was primarily used as an analytical framework. Public health research is increasingly highlighting the value in intersectionality frameworks for attempting to reduce inequalities. The lack of studies in our review suggests that theoretical and methodological advancements need to be made in order to enhance engagement with intersectionality as part of the health intervention development and assessment cycle.

Thank you, now changed.

Thank you, now changed.

Thank you, now changed.

Reviewer 3 Report

The manuscript considers highly actual issue of implementation of intersectionality in health research. However, the manuscript is very descriptive and does not contain any conclusions on how the consideration of intersectionality impacts the quality of interventional health research and any practical recommendations for implementation if intersectionality in the future research. The main limitation of the study was correctly identified by the authors. The review includes only studies mentioning intersectionality or related terms but neglects the vast majority of the studies including intersectionality factors in multivariate statistical analysis without using these terms.

Minor comments:

-   “intersectionality theory” and “intersectionality theories” are used interchangeable. Please describe more precisely what do you refer to.

-         -  Please discuss the exclusion criterion “Not high-income country”. Why this limitation? 

-          Please describe more precisely the inclusion criterion “use of intersectionality”. What degree of intersectionality implementation was needed?

Author Response

Thank you for this comment. Our scoping review set out to explore the use of intersectionality in interventional health research as well as analytical approaches. While we accept the comment the review is quite descriptive, we do not feel with the data we retrieved that it would be appropriate to comment on how intersectionality impacts on the quality of the research. Our objectives and chosen methodology of a scoping review, also directed our approach in being more descriptive than evaluative. We have now noted this specifically:

Page 12, line 34: Our scoping review intended to map out the extent to which intersectionality is being used in interventional health research and highlight some examples of its use. An extension of this work could assess whether and how intersectionality impacts on the quality of the interventional health research comparative to the use of other theoretical and analytical frameworks.

We appreciate this comment that it would be beneficial to highlight recommendations for implementing intersectionality in future research. We have revised some text where we direct readers to practical examples:

Page 11, line 44. While our small sample size limits what we can recommend in terms of intersectionality and intervention research, we would suggest that incorporating intersectionality in the design of interventions (not as a post-intervention analysis), would strengthen the equity focus. As fewer studies used intersectionality in this way, our review conveys that there is still some way to go before this becomes more commonplace. We would direct researchers to resources such as Hankivsky et al’s  (46) primer and Heard et al’s paper (19), which both provide case study examples of innovation using intersectionality frameworks in public health research. Additionally, the ‘For-Equity’ website also provides tools and resources related to intersecting inequalities to enable researchers to think about this when developing and evaluating interventions 47.

Thank you for this comment.

We thank the reviewer for this comment and have changed this to ‘intersectionality theory’ throughout.

Please see response to the Editor above, in the third comment row, which addresses this important comment.

We have added the following text clarify this at line page 5, line 12.

We remained open to a range of uses of intersectionality, expecting heterogeneity between studies in both the degree to which it was employed as either a framework for design or analysis, as well as the way it was interpreted and applied. Therefore, there may be studies that mention that they use intersectionality as a lens in their early discussions during the design of the intervention as well as studies that employ intersectionality as a guiding framework throughout the whole of the design and evaluation process – and both these would be included under our criteria.

Reviewer 4 Report

The article is well-written and organized. 

In a few places, references seem to be missing, so a light edit to correct this is necessary. For example, lines 72 and 73, Bowleg (unless I am missing something), the citation is missing. 

I do not think it is appropriate for a journal article to have review questions (106-114) and inclusion criteria (134-146) numbered out. I think it is much more acceptable to have them written out and a little explanation for each. I believe this helps the reader and strengthens the article. 

Author Response

Thank you. We have made the relevant changes to the references.

We have changed both the review objectives and inclusion criteria to reflect this comment:

Page 5, line 7. Our scoping review set out to explore the use of intersectionality theory and/or frameworks when designing or implementing public health interventions. We also aimed to identify intersectionality-based analytical approaches to examining the impact of interventions on health inequalities.

Page 5, line 44. Inclusion criteria

To be included in our review, studies should address intersectionality when designing, implementing and/or evaluating interventions. We remained open to a range of uses of intersectionality, expecting heterogeneity between studies in both the degree to which it was employed as either a framework for design or analysis, as well as the way it was interpreted and applied. Therefore, there may be studies that mention that they use intersectionality as a lens in their early discussions during the design of the intervention as well as studies that employ intersectionality as a guiding framework throughout the whole of the design and evaluation process – and both these would be included under our criteria.  All study types were accepted, including quantitative, qualitative and mixed method approaches. We limited studies to being high-income country-based. We limited our inclusion criteria to high income countries for a few reasons. *Justification for high income countries which appears in comment above*

To be included, we restricted studies to being health-related studies (i.e. measuring or analysing any indicator or topic related to physical or mental health of populations, as well as health system related outcomes), as opposed to educational or social outcomes, providing original results.  

Exclusion criteria

We excluded studies that were commentaries, editorials, book reviews or studies exclusively focused on educational, sociological, or judicial issues.